# Thermodynamic Limits and Optimality of Microbial Growth

**DOI:** 10.3390/e22030277

**Published:** 2020-02-28

**Authors:** Nima P. Saadat, Tim Nies, Yvan Rousset, Oliver Ebenhöh

**Affiliations:** 1Institute of Quantitative and Theoretical Biology, Heinrich-Heine-Universität Düsseldorf, Universitätsstraße 1, 40225 Düsseldorf, Germany; nima.saadat@hhu.de (N.P.S.); tim.nies@hhu.de (T.N.); yvan.rousset@hhu.de (Y.R.); 2Cluster of Excellence on Plant Sciences (CEPLAS), Heinrich-Heine University, Universitätsstrasse 1, 40225 Düsseldorf, Germany

**Keywords:** energy, entropy, anabolism, catabolism, microbial cultures, biotechnology

## Abstract

Understanding microbial growth with the use of mathematical models has a long history that dates back to the pioneering work of Jacques Monod in the 1940s. Monod’s famous growth law expressed microbial growth rate as a simple function of the limiting nutrient concentration. However, to explain growth laws from underlying principles is extremely challenging. In the second half of the 20th century, numerous experimental approaches aimed at precisely measuring heat production during microbial growth to determine the entropy balance in a growing cell and to quantify the exported entropy. This has led to the development of thermodynamic theories of microbial growth, which have generated fundamental understanding and identified the principal limitations of the growth process. Although these approaches ignored metabolic details and instead considered microbial metabolism as a black box, modern theories heavily rely on genomic resources to describe and model metabolism in great detail to explain microbial growth. Interestingly, however, thermodynamic constraints are often included in modern modeling approaches only in a rather superficial fashion, and it appears that recent modeling approaches and classical theories are rather disconnected fields. To stimulate a closer interaction between these fields, we here review various theoretical approaches that aim at describing microbial growth based on thermodynamics and outline the resulting thermodynamic limits and optimality principles. We start with classical black box models of cellular growth, and continue with recent metabolic modeling approaches that include thermodynamics, before we place these models in the context of fundamental considerations based on non-equilibrium statistical mechanics. We conclude by identifying conceptual overlaps between the fields and suggest how the various types of theories and models can be integrated. We outline how concepts from one approach may help to inform or constrain another, and we demonstrate how genome-scale models can be used to infer key black box parameters, such as the energy of formation or the degree of reduction of biomass. Such integration will allow understanding to what extent microbes can be viewed as thermodynamic machines, and how close they operate to theoretical optima.

## 1. Introduction

Life is certainly one of the greatest wonders on earth. The ability to grow and reproduce distinguishes living systems fundamentally from inanimate objects and engineered devices. However, despite their extraordinary complexity, living organisms obey fundamental physical laws as any other system. Therefore, it is not surprising that generations of scientists have studied life, and in particular microbial growth, using concepts and theories from physics. Whether microbes can be considered to be sophisticated biochemical machines is a deeply philosophical and highly relevant question [1]. We do not wish to enter this discourse here, and rather adopt a pragmatic point of view. Certainly, every microbe uses available free energy gradients (either from chemicals in the environment, or—in the case of photosynthetic organisms—from sunlight) to convert chemicals from the environment into a copy of itself. Thus, from a thermodynamic viewpoint, microbes are technically machines that convert free energy gradients into chemical work, which is used to construct a copy of themselves. So why should not concepts, principles, and formulas from thermodynamics be applicable also to microbial growth? In fact, it was exactly this mindset that led generations of scientists to develop excellent thermodynamic theories of microbial growth in the second half of the 20th century [2]. Notably, a major motivation to study microbial growth were biotechnological applications [3]. This is reflected by the fact that many models ignored details of microbial metabolism, but instead considered only the overall conversion of chemicals by a cell, which is described by a macrochemical equation [4]. This approach became known as black box modeling of microbial growth [5,6,7]. This view fits well with the available experimental technologies at that time [8,9]: In bioreactors and fermenters, fluxes of matter and energy into and out of the cells could be precisely measured, and, together with observed growth rates, were ideal to challenge the developed theories [2,10,11,12].

With the advent of high-throughput technologies, we have now an increasingly detailed picture of the internal processes within a microbe. Especially metabolism, the biochemical networks converting chemicals into chemical energy and biomass, is described with increasing detail. Consequently, new modeling techniques and theories have emerged to describe and investigate genome-scale metabolic networks [13]. These modern approaches now form a separate research field, which unfortunately appears to have little connection to existing thermodynamic theories of life. Scientists developing genome-scale network models and corresponding analysis techniques often seem to be unaware of the fundamental thermodynamic theories and thus ignore an invaluable heritage left behind by generations of great minds. And yet, detailed metabolic models can provide an unprecedented opportunity to peek inside the black box models of microbial metabolism.

The purpose of this review is to provide a summary of essential developments of theory building, both from the times before high-throughput technologies and after, as well as non-equilibrium thermodynamics to describe self-replication. We outline and discuss thermodynamic limitations and optimality principles that are revealed by different research approaches, before we develop links and connections between modern modeling approaches and thermodynamic theories of microbial growth. We aim to outline strategies how these promising but only loosely connected research fields can be integrated to mutually benefit from each other, and to offer opportunities to deeper understand microbial growth and discover fundamental principles of life.

## 2. Historical Overview

The economy of energy flow in living matter is a fascinating and complex topic that is and was investigated by generations of great scientists. Starting with the pioneering work of Jacques Monod [14,15,16], this section aims to give a concise overview of the historical developments in the treatment of thermodynamic aspects in models describing microbial growth. By including not only pure theoretical work but also past biotechnological examples, this section illustrates the motivation of the researchers to focus on energetic aspects of microbial cultures, thereby uncovering numerous interesting relationships.

Starting with Monod in the 1940s [15], who defined the biomass yield factor as the amount of biomass formed divided by the amount of limiting nutrient used (**Y**), the quantitative description of the production capabilities in microbial cultures evolved into a whole theory. The introduction of a simple hyperbolic relationship that connects the amount of limiting resources in the environment to growth rates of organisms opened the door for many different modeling approaches. Even today, the Monod growth law together with modern mathematical tools forms the basis for sophisticated theories of microbial growth. Although the work of Jacques Monod is undoubtedly one of the major achievements in microbiology of the 20th century, it is worth mentioning that others have proposed growth laws with slightly different functional dependencies of the growth rate on substrate availability, see for instance the work of Blackman and Tessier [17,18]. As Esener et al. show, all three models (Monod, Blackman, Tessier) can realistically describe experimental data of batch cultures [19].

In 1956 Herbert et al. introduced the concept of continuous cultures [20]. In contrast to batch cultivation, continuous cultures can be operated continuously for very long times, and thus presented a considerable advance in biotechnology. Moreover, the growth rate can be controlled by adjusting the dilution rate of the continuous cultivation system. Besides describing these systems experimentally, Herbert also developed theories for their quantitative description [20]. In 1958, Herbert realized [19,21] that the yield factor introduced in the 1940s by Monod is not constant but changes with the dilution rate. Both Herbert (1958) and Pirt (1965) tried to explain this behavior by introducing maintenance terms. The underlying assumption is that so-called maintenance functions, including turnover of cell material or maintenance of concentration or osmotic gradients, that are essential for the survival proceed with a rate that is independent on the growth rate [21,22]. Although Herbert [21] and Pirt [22] provide different explanations for these maintenance processes, Esener et al. have argued that both explanations can be used to essentially obtain the same results [19]. For a detailed discussion of different maintenance parameters and their measurement see [23].

With these theories in hand, the question appeared how essential parameters of microbial systems could be easily and relatively quickly estimated, given the diversity of organisms used in the laboratories or in biotechnology, especially regarding the limited information about the metabolism available at that time. The solution was the introduction of so-called black box models that only need input and output information, which can be readily obtained by controlling a bioreactor [3,5,7,8,9,19,24,25]. However, since the applicability of black box models heavily depends on energy and mass balance, a profound thermodynamic background is needed to understand and justify these approaches. Black box models were successfully applied to reveal many interesting relationships between energy, biomass and yield.

As Mayberry et al. stated in the 1960s, there was a considerable interest in investigating the proportionality between the yield and the converted energy [26]. One important contribution was made by Mayberry et al. in 1968, who observed that for bacteria growing on a single carbon compound that serves both as carbon and energy sources, the amount of produced dry weight (DW) of biomass per available electrons (av $e^{-}$) is relatively constant (3.14 gDW/av $e^{-}$) for a wide range of organic compounds. By redefining the concept of the "degree of reduction", which was first proposed by Gunsalus and Shuster [27], as "[number] of those electrons in a compound not involved in orbitals with oxygen [...]", and using the above mentioned observation, Mayberry et al. stressed the usefulness of counting available electrons for the analysis of growth data. A significant contribution was to introduce the degree of reduction in the treatment of energetic aspects in biology as a measurement of reducing power of a compound. In 1973 Minkevich and Eroshin [28] derived a formula for calculating the degrees of reduction (γ) of the electron donor, the biomass and metabolic products as well as a formula to obtain the dried biomass per oxygen using an overall metabolic reaction. However, great care must be taken when calculating and interpreting γ, since its derivation is dependent on the nitrogen source. Therefore, Roels later defined a generalized formula for the degree of reduction that can be used for any available nitrogen source [3].

By comparing numerous different organisms, carbon sources and literature, Minkevich and Eroshin [28] concluded that the degree of reduction of biomass ($\gamma_{b}$) for many microbial species situates around 4.2 (for comparison, glucose and acetate have a degree of reduction of 4 per carbon, while methanol has 6, and oxalic acid 1). In addition to this, they used old thermochemical knowledge from the 19th and early 20th century, namely that the molar combustion heats of organic compounds are nearly proportional to the consumed oxygen (known as Thornton’s rule [29]) to derive an efficiency parameter of growth (η), which specifies the energy fraction (contained in the substrate) transferred to biomass. This efficiency parameter sets a thermodynamic upper bound of the yield. Setting η=1, one obtains an upper bound of $Y<\frac{\gamma_{s}}{\gamma_{b}}$, where $\gamma_{b}$ denotes the degree of reduction of the biomass and $\gamma_{s}$ of the electron donor [24,28]. Nonetheless, this relationship is only valid if both the biomass and electron donor possess the same heat of combustion per available electrons. By studying different chemical species, Minkevich and Eroshin confirmed that the heat of combustion per gram-equivalent of oxygen consumed or per available electrons is for most organic substances around 26.94 kcal [28]. Similar values have been recalculated by numerous authors (for instance see [30,31]).

Later Erickson together with Minkevich and Eroshin introduced two more efficiency measurements: The fraction of the energy in an organic compound transferred to products ($\xi_{p}$) and the fraction that is evolved as heat (ϵ) during the growth process. Combined with the aforementioned energy fraction transferred to biomass (η), Erickson et al. developed simple mathematical tests to check for the consistency of various measured parameters, including energy requirements for growth and maintenance [24].

However, as Roels states in his excellent report, those parameters are based on a not entirely correct formulation of the second law of thermodynamics for open systems, because they assume that cellular growth always releases heat to the environment [3]. However, there are examples of microbes importing heat from the environment [8,11]. Roels provides a modified definition of maximum yield on substrate, $\omega_{f}$, that is consistent with the second law of thermodynamics. Moreover, he introduced two other definitions of maximum yield on substrate, one consistent with the heat transported to the environment, $\omega_{e}$, and another consistent with the oxygen transport, $\omega_{o}$, respectively. By showing that the values of $\omega_{f}$ and $\omega_{e}$ most often only deviate by 10%, he basically confirmed that Minkevich’s and Eroshin’s η was a useful parameter, at least under aerobic conditions.

Since $\omega_{f}$ is the maximum value of the yield that is consistent with the laws of thermodynamics, Roels defined the thermodynamic efficiency of a growth process as $\eta_{th}=Y_{sx}/\omega_{f}$, where $Y_{sx}$ is the yield of biomass (x) given a certain substrate (s). By calculating the values of $\eta_{th}$ for different organic compounds as energy source, he observed that highly reduced as well as highly oxidized substrates lead to a rather low efficiency. Roels concluded that substrates with a degree of reduction above 4.2 (the degree of reduction of biomass) contain enough energy to allow that all carbon could in principle be converted into biomass. In contrast to this, for compounds with a degree of reduction below 4.2, this is not possible due to energetic reasons.

How to incorporate these results into a mathematical model of bacterial growth was discussed by Esener in his report "Theory and Application of unstructured Models: Kinetic and Energetic Aspects" [19]. Together with his paper published in 1982, in which Esener et al. describe how to include aspects of varying biomass composition under a changing environment in a formal description of bacterial growth [32], his work is a good example of an early attempt to develop a consistent model explaining bacterial growth. These approaches are clearly related to modern approaches to explain bacterial growth laws (see [33,34]). For a very well written discussion comparing most of the efficiency measures discussed above (and much more), see [6,7].

In combination, these investigations and concepts of thermodynamic efficiency in bacterial growth provided the basis for further developments, including methods to estimate the energy and entropy of formation for biomass, which represents a parameter of fundamental importance for energetic calculations concerning life [30,31,35,36]. Particularly valuable were Battley’s contributions for estimating the Gibbs free energy of formation of biomass, $\Delta_{f}G_{b}$ (−65.10 kJ/C-mol, $\gamma_{b}$ = 4.998, N_2_ as nitrogen source), and the enthalpy of formation for biomass, $\Delta_{f}H_{b}$. By using the well-known relation of entropy, enthalpy and Gibbs free energy, ΔG=ΔH−TΔS, he even attempted to estimate the entropy of formation of biomass, but concluded that this method is too prone to errors, because it highly depends on the quality of the approximation of enthalpy and Gibbs free energy [31].

## 3. Recent Applications of Thermodynamics in Microbial Growth

The introduction of novel experimental techniques, often referred to as high-throughput ’omics’ technologies, around the beginning of the 21st century, allowed collecting extensive datasets and information about important molecular components of cells, including metabolites, RNA, DNA and protein levels. Falling cost and improved efficiency of genome sequencing gave rise to an ever-growing number of sequenced organisms [37]. Annotated genomes in turn resulted in comprehensive biochemical databases [38,39], which facilitated the construction of detailed metabolic models [40]. These models describe exactly those components which are inherently ignored in black box models, and thus provide a complementary theoretical approach to understand microbial growth. The question arises whether and how these approaches can be integrated.

Many analysis techniques to study large-scale metabolic network models, including flux-balance analysis (FBA) [41] and elementary flux mode analysis [42,43], are based on the assumption that metabolic fluxes are in a stationary state. This usually leads to a high-dimensional solution space and often those solutions are determined, which maximize a certain objective, such as growth rate or yield. However, without further constraints metabolic models may show thermodynamically infeasible flux solutions, such as ATP-generating cycles. Thus, in the traditional approach to calculate flux distributions *ad hoc* assumptions are made regarding the directions of the involved reactions. This information is often available in the original literature or included in biochemical databases such as KEGG or MetaCyc [38,39]. However, a binary distinction between reversible and irreversible does not fully account for the fact that the direction quantitatively depends on the equilibrium constant and the concentrations of the participating metabolites. Although the careful *ad hoc* assignment of reaction directions has led to successful applications to investigate uptake rates of carbon sources and byproduct secretion rates for several microorganisms [44,45,46,47], it was also realized that curating large-scale metabolic network models to ensure that no infeasible flux solutions can be produced is a time-consuming manual process [48].

Therefore, a main application of thermodynamics in metabolic models is to determine reaction directionalities. The Gibbs free energy of reaction ($\Delta_{R}G$) defines the direction in which a reaction proceeds (always in the direction of negative $\Delta_{R}G$) and is given by
(1)$$\Delta_{R}G=\sum_{i}n_{i}\Delta_{f}G_{i}^{0}+RT\sum_{i}n_{i}\;\ln\;c_{i},$$
where $n_{i}$ are the stoichiometric coefficients, $c_{i}$ the activities, and $\Delta_{f}G_{i}^{0}$ the respective standard Gibbs energies of formation, and the sum extends over all reactants of the reaction. Usually, the activities are approximated by the metabolite concentrations. Energies of formation can efficiently be computed using the group contribution method [49,50,51]. However, intracellular concentrations are much harder to obtain. To overcome this limitation, several different strategies have been developed. A heuristic approach based on network topology was introduced by Kümmel et al. [52], who described an algorithm assigning reaction directions based on the information on which side of the reaction energy-rich cofactors are found. To calculate thermodynamically feasible elementary flux modes in metabolic networks, Jol et al. [53] used a combination of quantitative metabolome data, flux variability analysis [54] and network-embedded thermodynamic analysis [55]. More recently, Peres et al. calculated thermodynamically feasible elementary flux modes by using equilibrium constants instead of internal metabolite concentrations [56]. A pragmatic strategy is to constrain reactions to physiologically feasible directions by estimating reasonable concentration ranges [57,58,59]. Beard et al. extended FBA by introducing energy balance analysis, which provides further constraints on flux bounds using estimated chemical potentials [60], resulting in thermodynamically feasible reaction directions for a genome-scale model of *Escherichia coli*. A more advanced strategy is to extend FBA to include metabolite concentrations as variables. Depending on the sign of Equation (Equation 1), the corresponding reaction is then constrained to a positive or negative flux. This results in a mixed-integer optimization problem, which ensures that the resulting flux distributions and metabolite concentrations are consistent with thermodynamic constraints [61]. In the approach by Hoppe et al. [61], the concentrations were constrained based on prior knowledge and experimental data.

A more recent technique to understand microbial growth based on optimality principles are resource allocation models [62,63,64], which treat microbial growth as an economic process. A cell has only a limited amount of available resources, such as nutrients, energy or space, and costs in terms of these resources are defined for the production of various cellular components, including macromolecules like enzymes and ribosomes. The principal idea of resource allocation models is to find an optimal strategy to allocate the available resources to maximize a certain output (usually the microbial growth rate). In fact, understanding microbial growth as an economic process, in which limited resources (such as macromolecules or space) have to be efficiently allocated, dates back to the pioneering work of Schaechter et al. [65,66] in the 1950s. Recently, comprehensive resource allocation models were proposed, which allow calculating and thus further understanding balances and trade-offs during microbial growth [62]. Molenaar et al. [67] observed that in contrast to flux-balance models where optimizing growth leads to a linear optimization problem, optimizing self-replication in resource balance models leads to highly complex behaviors even in simple self-replicator models. Employing thermodynamic flux-force relationships, Noor et al. showed that the logarithm of the ratio of forward and backward fluxes of a reaction is proportional to the Gibbs energy dissipation of the reaction [68], and therefore concluded that reactions with Gibbs energy dissipation close to zero require increased amounts of enzyme to facilitate a given net flux. In most resource allocation models, these thermodynamic contributions are not included. Extensive incorporation of thermodynamic principles has been applied by Niebel et al. [69] to analyze growth of *Saccharomyces cerevisiae* and *E. coli*, including thermodynamic constraints. In their work, it was suggested, based on experimental data, that an upper limit of the Gibbs energy dissipation rate exists. This assumption allows the formulation of an equation for the Gibbs energy balance, which constrains the sum of all Gibbs energy dissipation rates of all internal reactions to be equal to the exchange of Gibbs energy with the environment. The resulting models were used to predict oxygen uptake as well as byproduct and biomass production rates for different substrate uptake rates. Furthermore, some intracellular fluxes were predicted and then measured with ${}^{13}$C Metabolic Flux Analysis, which is an experimental approach to quantify metabolic fluxes using stable isotopes [70]. In their work it was observed that with increasing substrate uptake rates, the model shifted flux distributions from pathways with high dissipation rates, such as respiration, to pathways with lower dissipation rates, such as fermentation. Thus, they could provide a putative explanation, based on thermodynamic principles, for overflow metabolism phenomena, such as the Crabtree and Warburg effects, which traditional flux-balance models are unable to explain. Their approach, while certainly increasing thermodynamic rigor and improving the predictive capabilities, comes at the cost of increased computational complexity. To find flux distributions that maximize growth, the authors used a mixed-integer non-linear program.

However, it is interesting to note that the strict upper limit of Gibbs energy dissipation can be questioned. Older studies [71,72] showed that oxygen consumption remains constant for growth rates above a certain threshold, and that the cells additionally ferment glucose, which would lead to a further increase in Gibbs energy dissipation.

## 4. Thermodynamic Approaches to Self-Replication

The approaches discussed above employ classical thermodynamics to study microbial growth. Because the process of self-replication must obey fundamental physical laws, it should in principle be possible to describe it with concepts from statistical thermodynamics. In the 1990s, the development of the so-called fluctuation theorems [73,74,75,76] constituted a major advance in non-equilibrium statistical physics. Basically, they represent a generalization of the second law of thermodynamics. They link the probability to observe an entropy increase σ during a time interval τ to the probability to observe an entropy decrease by the same amount in the same time. This class of theorems can be generally expressed by
(2)$$\frac{P(+\sigma )}{P(-\sigma )}=\exp\left(\sigma \tau \right),$$
where P(+σ) and P(−σ) denote the probabilities to observe an entropy increase or decrease by σ during time τ, respectively. More recently, J. L. England [77] has proposed to apply this approach to self-replicating systems. Obviously, replication is a highly irreversible biological process and can be described in the language of statistical physics as a system that goes from a macrostate **I** (a single cell and the substrates in the surrounding medium), to a macrostate **II** (two daughter cells and the substrates), where each macrostate corresponds to an extremely large number of microstates. England’s reasoning starts from the fact that particles obey classical mechanics at the microscopic scale, and therefore follow a reversible dynamic. This allows quantifying the reversibility of a microscopic transition by the associated change in entropy. Applying these microscopic considerations to the macroscopic scale, the author obtains a generalization of the second law of thermodynamics for macroscopic irreversible biological processes.

While the classical second law of thermodynamics states that the increase of entropy of a closed system is always positive and obeys the inequality
(3)$$\Delta Q_{\mathrm{ex}}+T\Delta S_{\mathrm{int}}\geq 0,$$
where $\Delta Q_{\mathrm{ex}}$ is the amount of heat exchanged with the environment and $\Delta S_{\mathrm{int}}$ the internal entropy increase of the system, England’s derivation adds a new term to this relation,
(4)$$\frac{\Delta Q_{\mathrm{ex}}}{T}+\Delta S_{\mathrm{int}}+\ln\frac{\pi (\mathrm{II}\to I)}{\pi (I\to \mathrm{II})}\geq 0,$$
where π(I→II) (respectively π(II→I)) stands for the probability that the system evolves from macrostate **I** to macrostate **II** (respectively from **II** to **I**). When a macroscopic transition is irreversible (π(II→I)≪π(I→II)), the logarithm becomes negative, increasing the lower bounds for heat dissipation and entropy increase.

Equation (Equation 4) allows a closer look at self-replication. England applies this relation to a population of exponentially growing cells. He denotes the growth rate as *g* and the reverse rate (highly unlikely to happen) as δ. It allows expression of π(I→II) as gdt and π(II→I) as δdt. Hence, Equation (Equation 4) becomes
(5)$$\frac{\Delta q}{T}+\Delta s_{\mathrm{int}}\geq \ln\left(g/\delta \right),$$
where Δq and $\Delta s_{\mathrm{int}}$ are the respective intensive quantities. From this relation, one can see that the maximum duplication rate $g_{\max}$ is obtained when the right and left terms are equal. Therefore,
(6)$$g_{\max}-\delta =\delta \left(e^{\Delta q/T+\Delta s_{\mathrm{int}}}-1\right).$$Because the net growth rate is (g−δ), this last relation is of particular interest. The right-hand term shows the dependency on three quantities. The net maximal growth rate will increase with heat dissipation, Δq, internal entropy change, $\Delta s_{\mathrm{int}}$, and the rate at which the reverse process would occur, δ. The author emphasizes an interesting property: for identical entropy changes and decay rates, a replicator that dissipates more heat compared to another will have a higher maximal growth rate. A second particularly interesting aspect is the dependency of the net maximal growth rate on δ and $\Delta s_{\mathrm{int}}$. Low degree of organization and low stability make a replicator more competitive.

Application of Equation (Equation 4) to any self-replicator needs an estimation of the rates *g* and δ. England considers two examples, a self-replicating RNA and a replicating *E. coli* cell. For the former, RNA replication is considered to consist of a single ligation, which allows the author to estimate the decay rate δ from the half-life time of a phosphodiester bond and a duplication rate *g* from the RNA doubling time. Hence, he found a minimum limit of heat dissipation of $7\;\mathrm{kcal}\cdot {\mathrm{mol}}^{-1}$ for an RNA self-replicator, while the actual heat dissipation for the ligation reaction is measured experimentally as $10\;\mathrm{kcal}\cdot {\mathrm{mol}}^{-1}$. England argues that for the same reaction occurring for DNA, one finds a lower limit for heat dissipation of $16\;\mathrm{kcal}\cdot {\mathrm{mol}}^{-1}$, because of a considerably longer half-life of DNA compared to RNA. Therefore, such a ligation for DNA is thermodynamically not possible.

The same approach is applied to bacterial cell division. England considers a system with a single bacterium at constant temperature *T* in a rich medium. The system is initially in macrostate **I** (a single bacterium and the substrate), and will evolve through cell division to macrostate **II** (two bacteria and the substrate). Application of Equation (Equation 4) to this system provides a lower limit of heat dissipation for cell replication. This amount is six times lower than what was experimentally observed for *E. coli*, which is, according to England, surprisingly close to the thermodynamic limit.

In a very clearly described theoretical framework, Piñero et al. [78] have extended this approach to different types of self-replicating systems. They found specific thermodynamic constraints for each replicator type.

## 5. Combining Black Box Techniques with Modern Genome-Scale Approaches

In this section, we attempt to provide examples on how to integrate classical black box model techniques with modern genome-scale modeling approaches. Integrating thermodynamics into metabolic networks is currently one of the most fascinating and difficult challenges in the field of metabolic pathway analysis. However, methods and concepts developed for black box models are rarely applied in modern theoretical research. Highly relevant quantities are the Gibbs energies of reactions ($\Delta_{R}G$). To determine these requires knowledge of the Gibbs energies of formation ($\Delta_{f}G$) of the participating compounds. In the context of a black box model, only overall reactions are considered, in which nutrients are converted into biomass and byproducts. Therefore, calculating the corresponding Gibbs energy of reactions depends on the Gibbs energy of formation of the biomass ($\Delta_{f}G_{b}$). This quantity is directly accessible using, for instance, a technique described by Battley in 1993 [31] (see historical overview in Section 2).

To approximate the value of $\Delta_{f}G_{b}$, Battley considers a hypothetical combustion reaction of the biomass into fully oxidized components. For an experimentally determined elemental composition of the biomass of CH_1.595_O_0.374_N_0.263_P_0.023_S_0.006_ this reads
CH_1.595_O_0.374_N_0.263_P_0.023_S_0.006_ + 1.251 O_2_ + 0.012 KOH → CO_2_ + 0.131N_2_ + 0.006 P_4_O_10_ + 0.006 K_2_SO_4_ + 0.803 H_2_O.
Employing now the formula to calculate the degree of reduction when N_2_ is assumed to be the nitrogen source,
(7)$$\gamma =4n_{C}+n_{H}-2n_{O}-0n_{N}+5n_{P}+6n_{S}-n_{e^{-}},$$
in combination with the relation between the energy of combustion for an organic compound and its degree of reduction (Battley assumes −107.90 kJ/av $e^{-}$), he was able to estimate the energy of formation for biomass as −65.10 kJ/C-mol (not including ions). The $n_{X}$ in equation (Equation 7) denotes the number of atoms of type *X* or charge ($n_{e^{-}}$), respectively. The energies of formation necessary for obtaining this value are listed in Table 1.

Because all necessary information for repeating this calculation are contained in curated genome-scale models, it is rather straight-forward to transfer this old technique to modern modeling approaches. To demonstrate this, we determine the $\Delta_{f}G_{b}$-values for all genome-scale models contained in the BiGG database [79], using the specified biomass reactions. Figure 1 shows the distribution of energies of formation as function of the degree of reduction.

A comparison with the reported values ($\Delta_{f}G_{b}=-65.10$kJ/C-mol and γ=4.998 – see historical overview in Section 2) reveals that for most models energies of formation and degrees of reduction of the biomass are in agreement with former theoretical approximations. However, there are some values that deviate drastically from the mean (compare, for instance, upper right corner in Figure 1). Possibly, the observed variation results from different biomass compositions that were assumed for the specific models and their particular research questions. However, this kind of calculation offers the opportunity to scrutinize the plausibility of model assumptions, in particular referring to the biomass functions. For example, an energy of formation of +200kJ/C-mol seems highly unlikely.

Another possibility for a straight-forward approach to combine thermodynamic concepts from black box models with genome-scale models is a separate analysis of anabolism and catabolism. To calculate properties of anabolism, such as those predicted by Battley in 1993 [31], from genome-scale models, we pursue the following approach: Genome-scale models from the BiGG database were modified in two steps. First, all reactions that can produce ATP are disabled by introducing a dummy compound representing "unusable" ATP. Second, two strictly coupled reactions are introduced that import ATP into the cytosol, and export ADP and orthophosphate with the same rate. The strict coupling of import and export ensures that only energy, but no matter is introduced into the system. Thus, the modified model is unable to produce ATP from any carbon source and instead must use the imported ATP as energy source. Therefore, this modification separates anabolism from catabolism by simulating an external "ATP battery" providing the organism with external chemical energy, replacing the usual catabolic pathways.

These modified models were used to simulate anabolism separated from catabolism. In particular, we calculated the minimum amount of ATP required to incorporate one carbon atom from the nutrients into biomass and the minimum number of CO${}_{2}$ molecules that are released in this process. For this, the biomass production rate was fixed to the value obtained from the original model (in all cases the objective was maximization of the biomass production rate) and subsequently minimizing all carbohydrate import fluxes. The results are shown in Figure 2. Every point represents one model from the BiGG database. The x-axis displays the ratio of carbon dioxide produced by anabolism only to that produced by the original metabolism, in which anabolism and catabolism are coupled. The y-axis shows how many moles ATP are minimally required to incorporate one mole carbon atoms into the biomass. The large number of data points sharing the same anabolism versus metabolism ratio of released CO${}_{2}$ can be explained by the fact that a large proportion of models available in the BiGG database are for *E. coli*, and use the same biomass definition. For all *E. coli* models except the "core model", the required ATP per biomass carbon is between 2 and 3.5. Interestingly, the anabolism versus metabolism ratio of released carbon dioxide for the *E. coli* models is very close to the ratio predicted by Battley [31] (indicated by the dotted black line).

To illustrate how different modeling approaches can yield complementary interpretations of the same phenomenon, we consider in the following a simple thermodynamic black box model describing overflow metabolism in analogy to the detailed metabolic model developed by Niebel et al. [69]. The model is based on three assumptions. Assumption 1 states that growing cells can operate in two catabolic modes, which here correspond to fermentation and respiration, or a combination thereof. These catabolic reactions provide the required thermodynamic driving force for the overall metabolism. Assumption 2 states that the required Gibbs energy to convert one C-mol into biomass (the thermodynamic driving force) is dependent on the catabolic mode, and that fermentative growth requires less Gibbs energy per C-mol biomass than respiratory growth. This difference is motivated by numerous experimental data, which are summarized by Heijnen and van Dijken [6]. Finally, assumption 3 is identical to that made by Niebel et al. [69], stating that the total Gibbs energy dissipation rate is limited. These assumptions lead to two global constraints for cellular metabolism, one in the form of an energy balance, the other as an upper limit of the Gibbs energy dissipation rate (see Appendix B). If one further assumes that within the bounds defined by the constraints the cell uses that catabolic mode minimizing the total carbon uptake rate, short calculations (see Appendix B) lead to the results depicted in Figure 3.

Remarkably, these results are in very good quantitative agreement with experimental data (cf. Figure 2 in [69]). The virtue of the black box model is its extreme simplicity, which offers explanations for the observed behavior. The critical growth rate ($\lambda_{\mathrm{crit}}\approx 0.33$/h in Figure 3a) is reached when pure respiratory growth dissipates the maximal Gibbs energy per time. When this critical growth rate is exceeded, metabolism gradually shifts towards fermentation, because fermentation requires less Gibbs energy per C-mol biomass generated. Thus, higher growth rates can be obtained at the expense of a lower yield (Figure 3d), which is also manifested by the sharp increase in carbon uptake rate (Figure 3d). Naturally, a black box model is unable to explain or predict internal flux distributions. One of the key predictions of the detailed constraint-based model presented in [69] entails that in the limit of pure fermentative growth approximately 40% of the Gibbs energy is dissipated by only two enzymes, pyruvate decarboxylase and pyruvate kinase.

These considerations demonstrate the strengths and limitations of the different modeling approaches and outline how a combined analysis could considerably improve our understanding. Whereas black box models provide simple explanations for the observed growth behavior because of the model assumptions, detailed metabolic models allow investigating how the behavior is realized internally. An unsolved question is, for example, why fermentative growth requires less Gibbs energy for the same amount of biomass production when compared to respiratory growth. The detailed model could be used to examine the underlying mechanisms and pathways that are responsible for this experimentally confirmed observation. The models discussed here explain the growth behavior by assuming a limitation of the Gibbs energy dissipation. However, alternative explanations, such as limitations of heat production or cellular resources [62,80,81], are also conceivable. These and other hypotheses could be tested in a combined approach employing black box and detailed metabolic models. Thus, direct consequences of the assumptions and global limitations can directly be compared with predicted internal flux arrangements, paving the path towards finding mechanistic explanations for the complex emergent behavior of microbial growth (Appendix A).

## 6. Outlook and Conclusion

Increasing fundamental understanding of microbial self-replication and its limitations, thermodynamic considerations and calculations, can give valuable insights and expand the current knowledge about microbial growth. When attempting to answer the question whether prokaryotic organisms are thermodynamically optimized self-replicators, one is immediately confronted with several fundamental problems. There does not seem to be an agreement of what exactly "thermodynamic optimality" means. The black box models, discussed in Section 2, which arose in the 20th century focused on estimations of microbial growth efficiency parameters by only considering the exchange of matter and energy between cells and their environment. The parameters for aerobic microbial growth efficiency, proposed by Minkevich, Eroshin and Erickson [24,28], incorporated the degree of reduction of the biomass, substrates and electron donors to estimate a thermodynamic upper bound for the yield. This efficiency of biomass yield can be considered an important interpretation of thermodynamic optimality.

The upsurge of high-throughput techniques in the early 21st century gave rise to genome-scale metabolic models, opening the "black boxes" of microbial metabolism. These models provide excellent tools to investigate metabolic flux distributions. Some genome-scale metabolic models consider thermodynamics to constrain intracellular fluxes, but rarely for finding constraints for self-replication and growth itself. The work of Niebel et al. [69], mentioned in Section 3, is one of the most comprehensive approaches how to employ thermodynamics for the investigation of microbial growth using genome-scale metabolic models. Based on experimental data, the authors proposed an upper limit of Gibbs free energy dissipation during growth. They supported their hypothesis by implementing additional thermodynamic constraints reflecting this upper limit of Gibbs free energy dissipation. A flux distribution optimizing growth, which obeys this upper limit can also be considered to be an interpretation of thermodynamic optimality. A redistribution of fluxes to fermentative pathways due to constraints by the upper limit of Gibbs energy dissipation described by Niebel et al. [69] only occurs at high glucose uptake rates in *E. coli* and *S. cerevisiae*. Such conditions of excess nutrient availability are rarely found in nature and therefore an evolution towards such thermodynamic optimality is questionable. However, a hypothetical environment of fluctuating periods of excess nutrient availability could provide a fitness increase for microbial organisms adapted to flux redistribution obeying the upper limit of Gibbs energy dissipation.

A promising complementary approach attempts to describe microbial self-replication by basic principles from physics. In stark contrast to genome-scale models, which heavily depend on high-throughput data and computational power, applying physical concepts requires only a minimum amount of data. The work by England, mentioned in Section 4, aims at understanding microbial growth from a thermodynamic perspective. By applying fluctuation theorems to the non-equilibrium process of self-replication, a generalization of the second law of thermodynamics for irreversible transitions between two macroscopic states has been derived. This allowed calculating a lower bound for the produced heat during self-replication as a function of the internal entropy, growth and decaying rates. A consequence of these calculations is that a self-replicating microbe that dissipates heat with a rate close to the thermodynamic minimum is optimal in the sense that energy loss is minimized. However, the maximal rate of self-replication increases with increased heat dissipation. The finding that the heat dissipation of *E. coli* is not far from the calculated minimum needed for self-replication hints at evolution towards thermodynamic efficiency. However, the calculations imply that replication rates are increased with higher internal entropy and an increased rate of spontaneous self-decay. Both properties are not commonly found in microbial organisms. These properties may be beneficial to increase growth rate from a thermodynamic perspective, but are probably disadvantageous regarding other evolutionary pressures.

Evidently, there is no simple and unique answer to the question whether microbial growth is a process operating near thermodynamic optimum. Although all three concepts described here are concerned with the same phenomenon, each of them represents a different perception and viewpoint on the thermodynamic optimality of microbial growth. In our opinion these three concepts, as different as they may be, host an enormous potential to complement each other into an extended understanding of thermodynamic limitations and optimality of microbial growth.

The minimum amount of heat dissipation, and the upper limit of Gibbs free energy dissipation define fundamental thermodynamic limitations of microbial growth. The lower bound is a consequence of the extended second law of thermodynamics: It is impossible to replicate with less dissipated heat. The upper bound is an empirical observation which so far does not have a theoretical explanation. It implies that there is a theoretical upper limit for microbial growth rates. In addition, black box models allow calculating upper bounds of the yield based on physical-chemical properties of substrate and biomass, and most importantly the degree of reduction. It is not yet clear how these three limitations can be considered simultaneously, whether they agree, and how combining them might result in a more confined thermodynamic space.

An interesting observation was made by von Stockar [11], who reviewed thermodynamic data on microbial growth. This review includes examples of microbes that import heat from their environment, and compensate this by exporting chemical entropy [8]. Other microbes can do the opposite and even reduce the chemical entropy of their environment, at the expense of increased heat dissipation [82].

After pointing out that by combining the different interpretations of thermodynamics of microbial growth their potential could be drastically enhanced, we suggest that attempts to merge different thermodynamic theories should be further developed. One example of such a combined approach is the analysis of genome-scale models using constraints and concepts from black box models of microbial metabolism. Section 5 shows that in a relatively simple fashion, the combination of genome-scale models and black box models provides promising strategies to further understand thermodynamics of microbial growth and metabolism. However, for incorporating black box model concepts, metabolic networks need to guarantee to obey fundamental conservation laws. For example, calculating the upper thermodynamic limit of the yield ($Y<\frac{\gamma_{s}}{\gamma_{b}}$), requires the determination of the degrees of reduction for the biomass and all substrates. This heavily depends on a fully mass- and charge-balanced metabolic network. During our analysis, we noticed that only a few models actually fulfil this condition. This is understandable if one considers that most models were constructed for a completely different purpose. Finding flux distributions that optimize the incorporation of carbon into biomass also works without a perfectly charge-balanced network, and even an incorrect mass balance, which only concerns hydrogen atoms, will not affect the results. Nonetheless, to be truly reusable, in particular for thermodynamic calculations, any genome-scale model that is published should adhere to these fundamental chemical principles. We hope that this review encourages further activities to integrate different thermodynamic concepts and motivates the introduction of stricter standards to ensure reusability of genome-scale metabolic models.

## Figures and Tables

**Figure 1 entropy-22-00277-f001:**
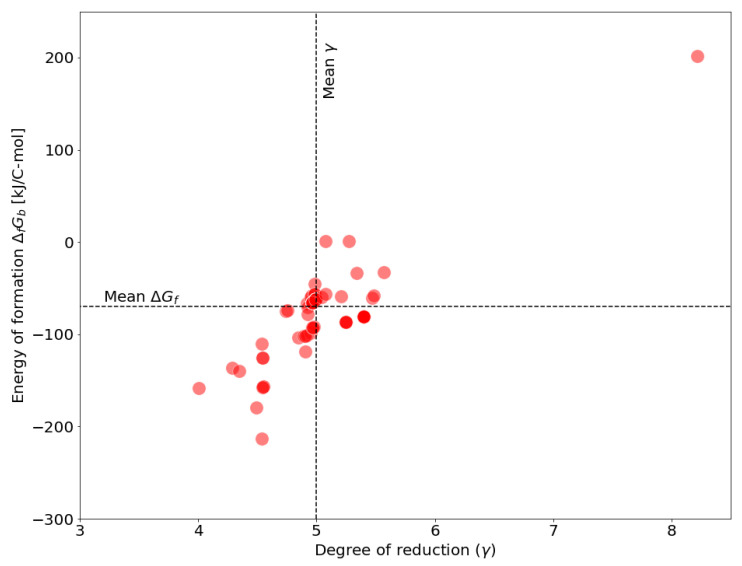
Energy of formation $\Delta_{f}G_{b}$ for biomass as encoded in genome-scale models of the BiGG database. Each point represents a biomass composition as described in the models. 85 models of the BiGG database containing 165 biomass functions were analyzed. Mean $\Delta_{f}G_{b}$ = −70.079 kJ/C-mol, Mean γ = 4.996.

**Figure 2 entropy-22-00277-f002:**
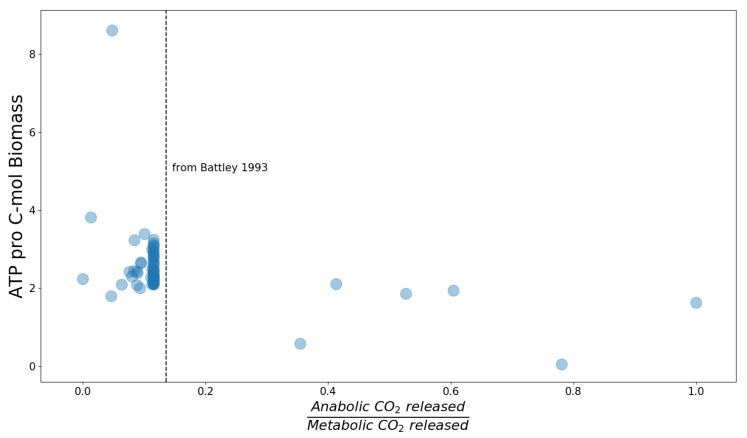
Anabolic properties of genome-scale models of the BiGG database. The y-axis indicates the minimum required amount of ATP per biomass carbon. The x-axis displays the ratio of carbon dioxide released by anabolism to carbon dioxide released by overall metabolism (including anabolism and catabolism).

**Figure 3 entropy-22-00277-f003:**
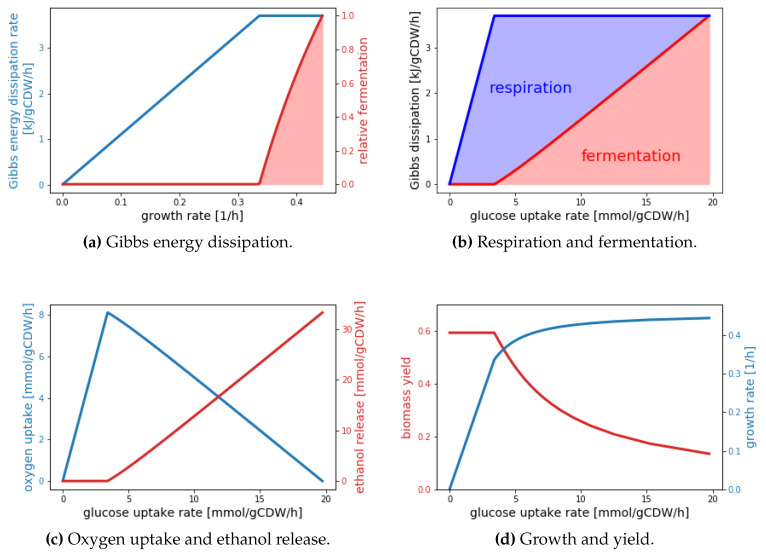
Black box model results for overflow metabolism.

**Table 1 entropy-22-00277-t001:** Standard free energy of formation for various organic compounds of interest necessary for estimating the energy of formation for biomass (see [31]).

Substance	Formula	$\Delta_{f}G_{i}^{0}$ [kJ/mol]
Oxygen	O_2_ (g)	0
Potassium hydroxide	KOH (c)	−379.11
Carbon dioxide	CO_2_ (g)	−394.36
Nitrogen	N_2_ (g)	0
Phosphorous decoxide	P_4_O_10_ (c)	−2697.84
Potassium sulfate	K_2_SO_4_ (c)	−1321.43
Water	H_2_O (lq)	−237.18

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
