# Peer review of "Thermodynamic Limits and Optimality of Microbial Growth"

_entropy, 2020, doi:10.3390/e22030277_

Round 1
Reviewer 1 Report
The authors have improved the paper very much.
The paper contains adequate references to the previous works and may be useful to specialists in the field.
However, it is unclear whether the paper provides accurate and useful information to the broad audience of Entropy. The self-contained aspect of the article seems to be somewhat lacking.
Reviewer 2 Report
I only have some minor final suggestions for the writing, as follow:
Line 72. “strategies about how”
Line 83. “purely”
Lines 112-113. “of growth rate”
Line 128. Is this citation #26?
Line 244. Reactants and products
Line 268. Internal or intracellular?
Line 533. “of whether”
Reviewer 3 Report
I now can suggest its publication. The editor can decide whether it is a review or a persepctive paper.
This manuscript is a resubmission of an earlier submission. The following is a list of the peer review reports and author responses from that submission.
Round 1
Reviewer 1 Report
The authors say that they "review various theoretical approaches that aim at describing microbial growth based on thermodynamic principles". However, I think that their review is very informal and insufficient to provide quantitative and knowledgeable information "to stimulate a closer interaction between related fields."
They "conclude by identifying conceptual overlap between the fields and suggest how the various types of theories and models can be integrated."
However, I think that their suggestion is very informal and insufficient to be published in a well-established scientific journal like Entropy.
I think that the paper does meet the reader's expectations that intuition about the relationship between the genome-scale approach and thermodynamic principles can be obtained.
Author Response
Reviewer 1:
The authors say that they "review various theoretical approaches that
aim at describing microbial growth based on thermodynamic
principles". However, I think that their review is very informal and
insufficient to provide quantitative and knowledgeable information "to
stimulate a closer interaction between related fields."
They "conclude by identifying conceptual overlap between the fields
and suggest how the various types of theories and models can be
integrated."
We have carefully revised the manuscript, paying attention to avoid a too informal style. Where possible, we have reformulated sentences and provided additional references supporting our statements.
Moreover, we have considerably expanded the section in which we outline strategies to stimulate a closer interaction between the fields. In addition to the quantitative calculations presented in our original submission, we have now also described a simple thermodynamic model that can complement the recently published genome-scale approach to study overflow metabolism (reference [69], Niebel et al.). Thus, we provide a quantitative and knowledgeable example how different modelling approaches give rise to complementary interpretations of the same phenomenon, thus stimulating a closer interaction between the field.
However, I think that their suggestion is very informal and
insufficient to be published in a well-established scientific journal
like Entropy.
We hope that our improvements and additions can convince the reviewer to revert his/her opinion.
I think that the paper does meet the reader's expectations that
intuition about the relationship between the genome-scale approach and
thermodynamic principles can be obtained.
We have considerably extended the section, in which we discuss the relationship between genome-scale models and thermodynamic principles. We have elaborated on numerous approaches how thermodynamic concepts are used to refine constraint-based metabolic models.
Reviewer 2 Report
This is an excellent, very enjoyable to read, and insightful review that aims at integrating classical thermodynamic theories of microbial growth with modern modelling that has emerged from the enormous information generated with the use of omics tools.
Perhaps some passages might be a bit too summarised and can be expanded to facilitate understanding e.g. lines 84-93 and maintenance requirements.
Although the manuscript is not difficult to understand, in some instances the writing can be improved. For example, lines 1-2 “To understand microbial growth with mathematical models has a long tradition that dates back to the pioneering work of Jacques Monod in the 1940s.” could read better as “Understanding microbial growth with the use of mathematical models has a long history that dates back to the pioneering work of Jacques Monod in the 1940s.” I pointed out other parts of the text in which the writing can be improved, but it is recommended to have the manuscript edited by a native English speaker.
Specific comments
Line 10. “the principal” instead of “principle”
Line 30. “is of course”
Line 31. “It is not surprise, therefore…”
Line 44. “its macroscopic behaviour” may read better
Line 44. Not clear to which theories are the authors referring to. The first theories?
Line 45. “fits well”
Line 50. Suggest “anabolism” instead of “metabolism”
Line 70. “section” instead of “part”
Line 82-83. It is not clear which three models are the authors referring to. The authors should expand a bit this passage to provide a more complete, even if brief, explanation
Line 95. It seems to me that this should be past tense “could”, as this is historical account
Line 102. “stated”
Line 105. “both as carbon and energy sources”
Line 106. “constant” instead of “stable”. Acronym DW should be defined as dry weight (although I recommend “dry matter”)
Line 110. “microbial species”?
Line 110. “situates” instead of “varies”
Line 132. “measurements”
Lines 140-141. Not clear if wf refers to “modified definition of maximum yield on substrate that are consistent with the second law”. Also, could it be “is” instead of “are” in line 141
Line 146. Perhaps a mathematical definition of wf would be of help at this point
Line 153. “was extensively”
Line 168. “to errors”
Line 173. “metabolites”
Lines 173-174. How about DNA?
Line 182. “microorganisms” instead of “microbial organisms”
Lines 184-187. Although it contains citations, it could be beneficial to expand and explain the limitation of flux balance analysis
Line 186. The authors previously define the acronym FBA in line 178 but spell it out here
There is a paragraph without lines between lines 187 and 188
Please insert comma: “Essentially, any enzymatic reaction…”
The authors may want to briefly explain that Gibbs energy changes are actually calculated based on activities rather than concentrations, although the latter approach is more commonly used
Line 196. Please briefly define molecular crowding
Lines 198-199. “focus” instead of “are focusing”
Line 218. Please define MFA
Again, there is a problem with line numbers after line 228
“during a time interval t”
Line 252. “of heat dissipation”
Line 259. “on how to integrate”
Line 259. Suggest using “classical” instead of “old”
Line 260. “Integrating” instead of “To integrate”
Lines 263-264. Should not this be singular i.e. “energy of reaction” and “energy of formation”?
Line 287. Please capitalize “Figure”
Line 304. “separated”
Line 320. “Increasing”
Line 321. Please insert comma after “calculations”
Line 324. “of what exactly”
Line 325. “which” instead of “that”
Line 328. Past tense “incorporated”
Line 332. Suggest “techniques” or “methods” instead of “data”
Line 335. “for finding” may read better
Lines 349-350. Not clear what “first principles” refer to. Perhaps basic principles?
Line 368. “each of them”
Lines 368-369. “the thermodynamic optimality”
Lines 370-371. I recommend the authors use the concepts of integration and generalization at this point
Line 375. “empirical”
Line 375. “which so far does not have a theoretical explanation”
Line 376. Do the authors mean theoretical upper limit? Unclear
Line 378. “and most importantly”
Line 387. Suggest “further developed”
Line 391. “for incorporating”
Line 397. “works”
Author Response
Reviewer 2:
This is an excellent, very enjoyable to read, and insightful review
that aims at integrating classical thermodynamic theories of microbial
growth with modern modelling that has emerged from the enormous
information generated with the use of omics tools.
We thank the reviewer for this encouraging remarks.
Perhaps some passages might be a bit too summarised and can be
expanded to facilitate understanding e.g. lines 84-93 and maintenance
requirements.
We have rewritten the respective passage, explaining the novelty of continuous cultures and the ideas behind the introduction of the maintenance terms.
The new text is now found in lines 86-98.
Although the manuscript is not difficult to understand, in some
instances the writing can be improved. For example, lines 1-2 “To
understand microbial growth with mathematical models has a long
tradition that dates back to the pioneering work of Jacques Monod in
the 1940s.” could read better as “Understanding microbial growth with
the use of mathematical models has a long history that dates back to
the pioneering work of Jacques Monod in the 1940s.” I pointed out
other parts of the text in which the writing can be improved, but it
is recommended to have the manuscript edited by a native English
speaker.
We have carefully checked the language and hope to have adequately improved the writing.
Specific comments
Line 10. “the principal” instead of “principle”
Line 30. “is of course”
Line 31. “It is not surprise, therefore…”
Line 44. “its macroscopic behaviour” may read better
Line 44. Not clear to which theories are the authors referring to. The first theories?
Line 45. “fits well”
Line 50. Suggest “anabolism” instead of “metabolism”
Line 70. “section” instead of “part”
Line 82-83. It is not clear which three models are the authors referring to. The authors should expand a bit this passage to provide a more complete, even if brief, explanation
Line 95. It seems to me that this should be past tense “could”, as this is historical account
Line 102. “stated”
Line 105. “both as carbon and energy sources”
Line 106. “constant” instead of “stable”. Acronym DW should be defined as dry weight (although I recommend “dry matter”)
Line 110. “microbial species”?
Line 110. “situates” instead of “varies”
Line 132. “measurements”
Lines 140-141. Not clear if wf refers to “modified definition of maximum yield on substrate that are consistent with the second law”. Also, could it be “is” instead of “are” in line 141
Line 146. Perhaps a mathematical definition of wf would be of help at this point
Line 153. “was extensively”
Line 168. “to errors”
Line 173. “metabolites”
Lines 173-174. How about DNA?
Line 182. “microorganisms” instead of “microbial organisms”
Lines 184-187. Although it contains citations, it could be beneficial to expand and explain the limitation of flux balance analysis
Line 186. The authors previously define the acronym FBA in line 178 but spell it out here
There is a paragraph without lines between lines 187 and 188
Please insert comma: “Essentially, any enzymatic reaction…”
The authors may want to briefly explain that Gibbs energy changes are actually calculated based on activities rather than concentrations, although the latter approach is more commonly used
Line 196. Please briefly define molecular crowding
Lines 198-199. “focus” instead of “are focusing”
Line 218. Please define MFA
Again, there is a problem with line numbers after line 228
“during a time interval t”
Line 252. “of heat dissipation”
Line 259. “on how to integrate”
Line 259. Suggest using “classical” instead of “old”
Line 260. “Integrating” instead of “To integrate”
Lines 263-264. Should not this be singular i.e. “energy of reaction” and “energy of formation”?
Line 287. Please capitalize “Figure”
Line 304. “separated”
Line 320. “Increasing”
Line 321. Please insert comma after “calculations”
Line 324. “of what exactly”
Line 325. “which” instead of “that”
Line 328. Past tense “incorporated”
Line 332. Suggest “techniques” or “methods” instead of “data”
Line 335. “for finding” may read better
Lines 349-350. Not clear what “first principles” refer to. Perhaps basic principles?
Line 368. “each of them”
Lines 368-369. “the thermodynamic optimality”
Lines 370-371. I recommend the authors use the concepts of integration and generalization at this point
Line 375. “empirical”
Line 375. “which so far does not have a theoretical explanation”
Line 376. Do the authors mean theoretical upper limit? Unclear
Line 378. “and most importantly”
Line 387. Suggest “further developed”
Line 391. “for incorporating”
Line 397. “works”
We sincerely thank the reviewer for this constructive and helpful list of suggestions. We have implemented all suggested changes in the revised manuscript.
Reviewer 3 Report
At first, I was excited about this paper because I thought that the main change needed would be to change the word "machines" to "organisms" throughout the manuscript.
But, there are a lot of typos and other corrections that need to be made. Line 2, delete the word "simple".
Line 24, change "machines" to "organisms."
Lines 30-31, the phrase "As such, life is nothing special..." could easily be argued with. Has it been definitely found elsewhere in the universe?
Lines 33, 37. Again, I believe that "machine" should be changed to "organism"
Line 44 regarding "limited knowledge of the detailed biochemical processes..." needs a citation.
Lines 52-58 need a lot of explaining and/or citations.
On the next line, "black box" which is italicized needs to be cited and defined. The use of "black box" continues throughout the manuscript.
On line 80, I don't understand why Monod is one of the major achievements in the life sciences of the 20th century. Please elaborate or supply a citation.
On line 92, there is a reference to "three models". These need to be described and/or cited.
On line 94, the term "batch cultivation" is introduced. This needs to be defined. Is this paper about a chemostat process? I am confused now.
The term "black box models" continues to be used.
At this point, I am finding so many issues on each page that I need to stop reading. I think that the paper should be rejected. The authors should be invited to do a major revision but the article will begin the review process at stage 1.
I wish my review was more positive.
Author Response
Reviewer 3:
At first, I was excited about this paper because I thought that the
main change needed would be to change the word "machines" to
"organisms" throughout the manuscript.
But, there are a lot of typos and other corrections that need to be
made. Line 2, delete the word "simple".
Line 24, change "machines" to "organisms."
We thank the reviewer for the comment, which helped us to realise that our usage of the term ‘machine’ might have caused misunderstandings. As a result, we have carefully rewritten in particular the introduction to make it clear how the term ‘machine’ should be understood. We explicitly mention that in a thermodynamic sense there are parallels between engineered machines and microbial cells. However, we reduced the usage of the term ‘machine’ to a minimum.
Lines 30-31, the phrase "As such, life is nothing special..." could
easily be argued with. Has it been definitely found elsewhere in the
universe?
We agree that this phrase was arguable and have removed it.
Lines 33, 37. Again, I believe that "machine" should be changed to
"organism"
See above.
Line 44 regarding "limited knowledge of the detailed biochemical
processes..." needs a citation.
We have changed the sentence to “Notably, a major motivation to study microbial growth were biotechnological applications.” and provided a reference to support our claim (Ref. [3], Roels).
Lines 52-58 need a lot of explaining and/or citations.
We have considerably rephrased the respective and the following paragraph. The new text now reads:
These modern approaches now form a separate research field, which unfortunately appears to have little connection to existing thermodynamic theories of life. Scientists developing genome-scale network models and corresponding analysis techniques often seem to be unaware of the fundamental thermodynamic theories and thus ignore an invaluable heritage left behind by generations of great minds. And yet, detailed metabolic models can provide an unprecedented opportunity to peek inside the black box models of microbial metabolism.
The purpose of this review is to provide a summary of essential developments of theory building, both from the times before high-throughput technologies and after, as well as non-equilibrium thermodynamics to describe self-replication. We outline and discuss thermodynamic limitations and optimality principles that are revealed by different research approaches, before we develop links and connections between modern modelling approaches and thermodynamic theories of microbial growth. We aim to outline strategies how these promising but only loosely connected research fields can be integrated to mutually benefit from each other, and to offer opportunities to deeper understand microbial growth and discover fundamental principles of life.
On the next line, "black box" which is italicized needs to be cited
and defined. The use of "black box" continues throughout the
manuscript.
We have included a number of references in which the term “black box” models was introduced. We think it is a standing expression in the field of biotechnology, and therefore continue to use it throughout the manuscript. However, we have added explanation how the term “black box” should be understood.
On line 80, I don't understand why Monod is one of the major
achievements in the life sciences of the 20th century. Please
elaborate or supply a citation.
We have changed “life sciences” to “microbiology”. We believe there is no doubt that Monod has made major contributions to shape this whole field.
On line 92, there is a reference to "three models". These need to be
described and/or cited.
These three models have been mentioned, alongside the references that were already present in the original submission.
On line 94, the term "batch cultivation" is introduced. This needs to
be defined. Is this paper about a chemostat process? I am confused
now.
We hope that the message of the paper is now clearer. We discuss the phenomenon of microbial growth from a thermodynamic perspective without restricting the view to chemostats or batch cultures only.
The term "black box models" continues to be used.
See above.
At this point, I am finding so many issues on each page that I need to
stop reading. I think that the paper should be rejected. The authors
should be invited to do a major revision but the article will begin
the review process at stage 1.
As stated above, we have thoroughly revised our manuscript and hope it is now more accessible.
I wish my review was more positive.
Reviewer 4 Report
The authors reviewed the history of thermodynamic theory and recent high-throughput technology development of microbes grows. Although the language is clear, the contents are poor. They separately mentioned two scientific branches and call for their integration, but the existing integration so far is rather few. It more deserves a perspective rather than a review paper. And also the title is strange. I donot think it indicated the most important contribution of the paper.
Author Response
Reviewer 4:
The authors reviewed the history of thermodynamic theory and recent
high-throughput technology development of microbes grows. Although the
language is clear, the contents are poor.
We are sorry about this negative impression our manuscript left. However, it would be helpful to support the statement and give scientific arguments why the reviewer considers the contents as poor.
They separately mentioned
two scientific branches and call for their integration, but the
existing integration so far is rather few.
First, we agree with the reviewer that in the current literature there are very few examples how the two scientific branches can be integrated. This is exactly one of our main motivations for writing this article. We understand that also the example calculations we have provided in the original submission was not satisfactory for the reviewer. During our careful revision of the manuscript, we have now also described a simple thermodynamic model that can complement the recently published genome-scale approach to study overflow metabolism (reference [69], Niebel et al.). Thus, we hope we now clearly outline how different modelling approaches give rise to complementary interpretations of the same phenomenon, thus stimulating a closer interaction between the field.
It more deserves a
perspective rather than a review paper.
We agree that our paper has aspects of both types of manuscripts. Indeed, it was our purpose to also give a perspective on how to integrate different scientific approaches in the future. However, in order to outline such strategies, it is a prerequisite that the respective fields are thoroughly reviewed. Thus, we still consider our article to be more Review than Perspective, but will leave the ultimate decision on that subject to the Editor.
And also the title is
strange. I donot think it indicated the most important contribution of
the paper.
We understand that the title might have not adequately reflected the main messages of the paper and changed it into “Thermodynamic limits and optimality of microbial growth”.